# Phytochemical Composition, Antioxidant and Antifungal Activity of *Thymus capitatus*, a Medicinal Plant Collected from Northern Morocco

**DOI:** 10.3390/antibiotics11050681

**Published:** 2022-05-18

**Authors:** Amina Benoutman, El Hadi Erbiai, Fatima Zahra Edderdaki, El Khalil Cherif, Rabah Saidi, Zouhaire Lamrani, Manuela Pintado, Eugénia Pinto, Joaquim C. G. Esteves da Silva, Abdelfettah Maouni

**Affiliations:** 1Biology, Environment, and Sustainable Development Laboratory, ENS, Abdelmalek Essaadi University, Tetouan 93000, Morocco; amina.benoutman@etu.uae.ac.ma (A.B.); elhadi.erbiai@etu.uae.ac.ma (E.H.E.); fatimaezzohra.eddardaki@etu.uae.ac.ma (F.Z.E.); r.saidi@uae.ac.ma (R.S.); zlamrani@uae.ac.ma (Z.L.); 2Chemistry Research Unit (CIQUP), Institute of Molecular Sciences (IMS), DGAOT, Faculty of Sciences, University of Porto, 4169-007 Porto, Portugal; c.elkhalil@uae.ac.ma (E.K.C.); jcsilva@fc.up.pt (J.C.G.E.d.S.); 3Institute for Systems and Robotics, Instituto Superior Técnico, University of Lisbon, 1049-001 Lisboa, Portugal; 4CBQF—Centro de Biotecnologia e Química Fina, Laboratório Associado, Escola Superior de Biotecnologia, Universidade Católica Portuguesa, 4169-005 Porto, Portugal; mpintado@porto.ucp.pt; 5Laboratory of Microbiology, Biological Sciences Department, Faculty of Pharmacy, University of Porto (FFUP), 4050-313 Porto, Portugal; epinto@ff.up.pt; 6CIIMAR-Interdisciplinary Center of Marine and Environmental Research, University of Porto, 4050-313 Porto, Portugal

**Keywords:** *Thymus* *capitatus*, biochemical composition, essential oil, antioxidant activity, genotoxicity test, antifungal activity

## Abstract

*Thymus capitatus* is a Mediterranean endemic plant commonly known as “Zaïtra” in northern Morocco. As *T. capitatus* is widely used in traditional medicine and food, this present work aims to investigate the chemical compositions and biological activities of the *T. capitatus* leaves essential oil (TcLEO), acetonic (TcLAE), and methanolic extract (TcLME). The spectrophotometric determination demonstrated that *T. capitatus* is a natural source rich in phenolic contents (TPC) and flavonoid contents (TFC) and that TcLME revealed the highest TPC and TFC than TcLAE and TcLEO. The LC-MS analysis of phenolic compounds showed that paraben acid was predominant in both TcLME and TcLAE, followed by cinnamic acid and *p*-hydroxybenzoic acid. GC-MS analysis of the TcLEO revealed the presence of a total of 10 compounds, which were predominated by carvacrol. The antioxidant activity by ORAC was observed to be significantly higher in TcLEO and TcLAE than in TcLME. All samples used to assess DNA degradation effectively prevented DNA oxidation and, at the same time, had a prooxidant effect. The genotoxicity test showed that the *T. capitatus* were devoid of any mutagenic activity. Concerning antifungal activity, all samples were able to inhibit the growth of all microorganisms tested at low concentrations. TcLAE showed higher activity than TcLME, and in general, dermatophytes were more susceptible, being *Microsporum canis* the most sensitive one. Overall, the results obtained from this study confirm the wide uses of *T. capitatus*. Furthermore, the finding results suggest that the *T. capitatus* essential oil and extracts can be highly useful for pharmaceutical industries.

## 1. Introduction

During the past few decades, medicinal and aromatic plants have become a source of natural compounds known for their biological activities (antioxidant, antibacterial, antifungal, anticancer, anti-inflammatory, etc.). However, there is a constant need to develop new natural compounds [1] with medicinal properties. Aromatic and medicinal plants have been the subject of many studies to analyze their chemical composition and evaluate their pharmaceutical properties. These plants were used for a long time in traditional medicine as antimicrobial agents [2,3]. Furthermore, for many thousands of years, essential oils (EOs) and plant extracts have been used for a wide variety of purposes, including perfumery, home care, personal care (perfumes and cosmetics), foods, and health-promoting agents for the treatment of different illnesses [4,5,6].

Fungal infections have been considered a real public health problem, particularly among patients with weakened immune systems (human immunodeficiency virus, lymphomas, transplants, and diabetes) [7]. The emergence of strains resistant to the current antifungals, associated with the modification of the clinical spectrum of pathogens and the overuse of conventional antifungal drugs, encouraged the research for alternative natural products such as organic extracts and EOs from medicinal plants [8,9,10].

Oxidative stress is involved in many diseases as a trigger or associated with complications [11]. Most of the diseases induced by oxidative stress appear with age, which leads to serious pathologies such as cardiovascular and neurodegenerative diseases, cancer, diabetes, metabolic syndrome, and digestive disease [12,13,14,15].

The degenerative process is due to reactive oxygen species (ROS) such as H_2_O_2_ and which are responsible for damaging lipids, proteins, and DNA and contribute to the development of many degenerative diseases [16], or even environmental exposure to prooxidant factors such as tobacco, alcohol, drugs, and pesticides [17]. Hence, antioxidants are recommended, and the most used are those of synthesis. Currently, the use of available synthetic antioxidant molecules is questioned due to the potential health risks and toxicity that they cause [18]. Furthermore, the need to develop alternative active compounds has become necessary due to the serious damage that synthetic antioxidants such as butylated hydroxyl toluene (BHT) and butylated hydroxyl anisole (BHA) leave to the consumer’s health and the environment [19,20,21]. Today, the study of plant antioxidant activities has become important [22]. Besides, the powerful antioxidant activities of the plants are due to the richness of polyphenols (phenolic acids, flavonoids, lignans, stilbenes), carotenoids, vitamins (vitamin E, vitamin C, and vitamin A), and trace elements (copper, manganese, selenium, and zinc) [23].

*Thymus capitatus* is a Mediterranean endemic plant belonging to the *Lamiaceae* family [24]. The *Lamiaceae* includes a large number of aromatic and medicinal plants (about 230 genera and 7100 species worldwide), many species from this family are known for their biological properties thanks to their richness in secondary metabolites and their uses in medicine and cosmetics [25,26,27,28]. The genus *Thymus* has been used in traditional medicine to treat several illnesses, including respiration inflammation, asthma, pneumonia, cancer, influenza, dermatitis, diabetes, gastrointestinal disorders, and heart diseases, and used as a sedative, analgesic, and flavoring and infusion for disinfection of the digestive tracts, also known for their antimicrobial, antiseptic, and antispasmodic activity [29,30,31,32,33,34,35]. This genus is represented in Morocco by 21 species, including 12 endemics. *T. capitatus* is commonly known in Morocco as “Zaïtra” and is widely used in traditional medicine, especially in the Arab region [36]. Nowadays, EOs and their components are gaining increasing interest because of their relatively safe status and exploitation for potential multi-purpose functional uses [37]; they can be (nano) encapsulated with zein to increase the shelf life of food by inhibiting fungus [38]. *T. capitatus* possess different biological effects such as antioxidant, antifungal, antibacterial, and antiviral [39,40,41,42,43,44,45,46].

This study aimed to determine the total phenolics and flavonoids contents, identification of phenolic compounds by LC-MS (Liquid Chromatography-Mass Spectrometry), determination of the chemical composition of the EO, and evaluation of biological assays, including the antioxidant activity ORAC (Oxygen Radical Absorbance Capacity) and DNA protection), antifungal activity, and genotoxicity test of methanolic, acetonic extracts, and EOs of *T. capitatus* plant leaves.

## 2. Materials and Methods

### 2.1. Standards, Reagents, and Medium

2-Diphenyl-1-picrylhydrazyl (DPPH) was purchased from Alfa Aesar (Ward Hill, MA, USA), 2,2′-azino-bis (3-ethylbenzothiazoline-6-sulfonic) acid (ABTS) was from (Sigma, Darmstadt, Germany). L-ascorbic acid, catechin, Folin–Ciocalteu phenol reagent, 6-hydroxy-2,5,7,8-tetramethylchroman-2-carboxylic acid (Trolox), sodium nitrite, sodium hydroxide, iron (III) chloride, alkane standard (C8-C37), and phenolic standards (gallic acid, protocatechuic acid, *p*-hydroxybenzoic acid, vanillic acid, *p*-coumaric acid, ferulic acid, syringic acid, paraben acid, and cinnamic acid) were purchased from Sigma-Aldrich (St. Louis, MO, USA). Aluminum chloride, ethyl acetate, and acetonitrile were obtained from Merck KGaA (Darmstadt, Germany). Fluorescein (C_20_H_12_O_5_), 2,2-Azobis (2- methylpropionamidine) dihydrochloride (AAPH), and 6-hydroxy-2,5,7,8-tetramethylchroman-2-carboxylic acid (Trolox), bromophenol blue, Tris EDTA (TE) buffer, phosphate buffer, phosphate-buffered saline (PBS), DMSO, and hydrogen peroxide were purchased (from Sigma, Darmstadt, Germany). Glycerol was obtained (from Thermo Fisher Scientific, Walthem, MA, USA), and Tris-Acetate EDTA buffer (TAE) (from BioRad, Hercules, CA, USA). GreenSafe Premium was obtained (from Nztech, Lisboa, Portugal). Acetone, n-hexane, and hexane were purchased from CABLO ERBA Reagent, S.A.S (Val de Reuil Cedex, France), Sabouraud dextrose agar (SDA) was purchased from Bio-Mèrieux (Marcy L’Etoile, France) and RPMI-1640 broth medium was purchased from Biochrom AG (Berlin, Germany). Methanol and all other chemicals and solvents were obtained from Honeywell (St. Muskegon, MI, USA).

### 2.2. Plant Materials

The *T. capitatus* plant studied was collected in May 2018 at Dar Ben Karrich *Tetraclinis* forest, a site of biological and ecological interest (SIBE) relevant to the province of Tetouan in northern Morocco (35°50′68″ N, 5° 42′79″ W, 220 m of altitude, thermo-Mediterranean vegetation level, a subhumid bioclimatic level at temperate winter, and siliceous substrate). The plant was identified according to the Manual for the determination of vascular plants (Practical Flora of Morocco) [47]. The *T. capitatus* leaves were weighed, cut into small pieces, air-dried naturally away from light and moisture on paper for 20 days at room temperature, then crushed to a powdered form (20 mesh) and stored in a dark glass bottle.

### 2.3. Organic Biomolecules Extracts and Essential Oil Preparation

The extracts preparation of *T. capitatus* leaves was performed by using two different solvents, methanol and acetone. The extraction was carried out according to the procedure used by Barros et al. [48] and modified by Erbiai et al. [49]. One gram of fine dried leaves powder was extracted by stirring with 20 mL of methanol and acetone at 25 °C at 150 rpm for 24 h and filtered through Whatman N° 4 Filter paper. The residue from the filtration was extracted again twice using the procedure described above. The combined extracts were evaporated at 40 °C to dryness using a rotary vacuum evaporator (Rotavapor^®^ R-210, BÜCHI, Flawil, Switzerland). Then, the dried extracts were weighed and stored at −81 °C for further use.

For EO preparation, 100 g of dried plant leaves were extracted with 500 mL of distilled water by the hydrodistillation method for 4 h using Clevenger-type apparatus [50]. The obtained EO was stored at 4 °C in dark bottles until further use. The extraction yield was calculated for each solvent used and obtained EO.

### 2.4. Spectrophotometric Determination of Total Phenolic and Flavonoid 

The total phenolic concentration was determined according to the *Folin-Ciocalteu* (FC) method used by Erbiai et al. [49]. One hundred µL of the plant extracts (1 mg/mL) or gallic acid at different proportional concentrations (500–15.62 µg/mL), 500 µL of the FC reagent, and 400 µL of Na_2_CO_3_ (7.5%) were incubated for 20 min at 40 °C. Then the absorbance was measured at 760 nm against distilled water as blank. All samples were performed in triplicate. The total phenol concentrations were expressed as mg of gallic acid equivalent (GAE)/g of dry extract (DE) or EO.

The concentration of the flavonoid was determined according to the method previously used by Erbiai et al. [49]. A mix of 250 µL of the plant extracts (1 mg/mL) or catechin (CE) at different proportional concentrations (500–15.62 µg/mL), 1250 µL of distilled water, and 75 µL of NaNO_2_ (5%) was incubated for 5 min. Then, 150 µL of AlCl_3_ (10%) was added to the mixture. After 6 min of incubation, 500 µL of NaOH (4%) and 275 µL of distilled water were added. The absorbance was measured at 510 nm using distilled water as a blank. All trials were performed in triplicate. The concentration of total flavonoid was expressed as mg of catechin equivalent (CE)/g of DE or EO.

### 2.5. Phenolic Compounds Analysis by LC-MS

The phenolic compounds analysis of *T. capitatus* leaves was performed following a procedure previously modified by the authors Erbiai et al. [51] by using the same conditions and HPLC equipment. Briefly, the phenolic extract was analyzed by liquid chromatography-mass spectrometry (LC-MS). Chromatographic separation was accomplished using an Acclaim™ 120 reverse phase C18 columns (3 µm 150 × 4.6 mm) thermostatted at 35 °C, and peaks were detected at 280 nm as the preferred wavelength. The mobile phase used was composed of 1% acetic acid and 100% acetonitrile. The identification of phenolic compounds in the samples was characterized according to their UV-Vis spectra and identified by their mass spectra and retention times in comparison with commercial standards. Quantification was made from the areas of the peaks recorded at 280 nm by comparison with calibration curves obtained from the standard of each compound. The results were expressed in µg per gram of dry extract (DE).

### 2.6. GC-MS Analysis

The EO of *T. capitatus* leaves was chemically analyzed using Gas Chromatography (GC) (Trace 1300 gas chromatography; Thermo Fisher Scientific, Waltham, MA, USA) linked to a mass spectrometry (MS) system (ISQ single quadrupole mass spectrometer; Thermo Fisher Scientific). The GC was equipped with a capillary column DB-5 (30 µm, 0.25 mm i.d, film thickness 0.25 µm) with a non-polar stationary phase (5% phenyl, 95% dimethylpolysiloxane). The column temperature was programmed from 50 to 350 °C at a rate of 5 °C/min. Helium was used as a carrier gas at a flow rate of 0.75 mL/min. The identification of components of leaves EO was based on retention indices (RI) relative to those of a homologous series of known standards of alkanes mixtures mixture (C_8_–C_20_ and C_21_–C_40_) and with the spectral data obtained from the databases of the National Institute Standard and Technology (NIST2014) [50].

### 2.7. Determination of Antioxidant Activity by ORAC

The ORAC (Oxygen Radical Absorbance Capacity) activity was performed according to the procedure employed by Contreras et al. [52] with small modifications. The reaction was performed in 75 mM phosphate buffer (pH 7.4), and the final reaction mixture was 200 Ll. Each well of Black polystyrene 96-well microplate containing 20 µL of Trolox standard (10 at 80 µM) or the sample at different concentrations (0.0125 at 0.0025 mg/mL), 120 µL of fluorescein, and 60 μL of AAPH.

This test was realized with a multi-detection plate reader (Synergy H1, Winooski, VT, USA) controlled by the Gen5 Biotek software version 3.04. The excitation wavelength was set at 485 nm and the emission wavelength at 528 nm. The fluorescence was recorded at intervals of 1 min for 80 min. AAPH and Trolox solutions were prepared daily, and fluorescein was diluted from a stock solution (1.17 mM) in 75 mM phosphate buffer (pH 7.4). All the reaction mixture was realized in duplicate, and at least three independent measurements were performed for each sample. The values of ORAC were expressed as µmol of Trolox equivalent (TE)/g of sample.

### 2.8. Assessment of DNA Degradation by Extract and Essential Oils of T. capitatus

According to the procedure employed by Silva et al. [53], two assays were used to evaluate the rate of degradation of calf thymus DNA by *T. capitatus* extracts and EO: prooxidant (PO) and antioxidant (AO), with and without FeCl_3_ (Iron) for two assays. The DNA solution was prepared in sterile distilled water at 0.25 mg/mL and stored in the dark for 7 days at 4 °C with homogenization every night. Stock solutions of extracts (methanolic and acetone) and EO were prepared in PBS-7.4 at 1 mg/mL. In both assays, the volumes of the different components were specified as follows: 200 µL of DNA with 10 µL of FeCl_3_ and without, 50 or 400 µL of H_2_O_2_ (7.01 M) only in the case of AO assay with iron and without, variable volume of sample (400, 300, 200, 100, 80, 60 and 40 µL) and PBS phosphate buffer adjusting the total volume to 1000 µL. The DNA solution adjusted by PBS, with iron and without, was used as the positive control (no degradation).

After 1 h of incubation in the dark at 37 °C, an agarose gel electrophoresis was run. For this purpose, each sample was mixed with loading buffer (1:4), and 10 µL were transferred to agarose solution (0.75%) gel prepared using TAE buffer supplemented with 0.03 µL/mL of GreenSafe Premium. Electrophoresis was then run for 1.25 h at 150 mV, gels were analyzed using a molecular imager (GelDOC XR), and the resulting image was processed using Image Lab Software v5.1 (BioRad, Hercules, CA, USA).

The band area for each positive control was manually defined to measure the band intensity and copied into each sample lane, maintaining the distance to the wells. The decrease in the band intensity was considered a result of the reduction of the amount of DNA present.

The results are specified as a percentage of inhibition of DNA band degradation for AO and the percentage of DNA degradation for PO assays, both calculated as expressed in the equations:$$\mathrm{Inhibition}\;\mathrm{of}\;\mathrm{DNA}\;\mathrm{degradation}\;\left(\%\right)=\left[\frac{I_{\mathrm{sample}}-{\;I}_{\mathrm{background}}}{I_{\mathrm{DNA}\;\mathrm{solution}\;}}\right]\times 100$$
$$\mathrm{DNA}\;\mathrm{degradation}\;\left(\%\right)=100-\left[\frac{I_{\mathrm{sample}}-{\;I}_{\mathrm{background}}}{I_{\mathrm{DNA}\;\mathrm{solution}\;}}\times 100\right]$$
where $I_{\mathrm{sample}}$ the intensity of each sample band, I_background_ is the intensity of the background, measured beside the control bands, and I_DNA solution_ is the intensity of the intact DNA solution. All tests were made in triplicate.

### 2.9. Genotoxicity Activity

The mutagenicity of the extracts and EO of *T. capitatus* were examined by the Ames test [54] at a concentration of 100 µg/mL, using the bacterium *Salmonella Typhimurium* TA98 with and without metabolic activation (S9). 2-aminoanthracen and daunomycin were used as a positive control, and DMSO 4% and water were used as a negative control.

### 2.10. Determination of the Antifungal Activity

Five fungal strains, American Type Culture Collection-ATCC and clinical strains, from the microbiology laboratory of the Faculty of Pharmacy, University of Porto, have been used in this study: *Trichophyton rubrum* FF5, *Epidermophyton floccosum* FF9, *Microsporum canis* FF1, *Candida albicans* ATCC 10231, and *Aspergillus fumigatus* ATCC 46645.

The antifungal activity of *T. capitatus* was performed by the broth microdilution method recommended by the Clinical and Laboratory Standard Institute-CLSI (M38-A2, filamentous fungi; M27-A3, yeasts) and previously described by Erbiai et al., [51]. Briefly, the dried extract (methanolic and acetonic) and EO were dissolved in dimethyl sulfoxide (DMSO) and serially diluted in RPMI 1640 medium to reach the concentrations to be tested (DMSO not more than 2% in the final solution prepared). 100 µL of each dilution were distributed in a 96-well plate. Inoculum of fungi was prepared in physiological water and standardized by spectrophotometric method for yeasts and by spores count for filamentous fungi. The prepared inoculum was diluted in RPMI to a final suspension of 10^3^ to 10^4^ colony-forming units (CFU)/mL, and 100 µL of the fungal suspension were added to each well. The plates were incubated for 48 h at 37 °C for *Candida* and *Aspergillus* and 7 days at 25 °C for dermatophytes (*T. rubrum, E. floccosum,* and *M. canis*). The minimum inhibitory concentration (MIC) was considered as the concentration that caused no visible growth, and the minimum fungicidal concentration (MFC) was determined by inoculating 10 µL taken from the wells with no turbidity in MIC determination into an SDA medium Petri dish. The MFC was the lowest concentration that totally inhibited the growth under the incubation conditions previously described. Controls: quality, using *Candida krusei* ATCC 6258 and performed with voriconazole; sterility, with RPMI-1640; and growth, with RPMI-1640 medium plus DMSO (1.0%) and fungi suspension.

### 2.11. Statistical Analysis

All tests were carried out in triplicate, and values were expressed as mean ± standard deviation (SD). The statistical significance of the data was made with a one-way analysis of variance (ANOVA), followed by posthoc Tukey’s multiple comparison test with a = 0.05 using GraphPad Prism 8.0.1 software (San Diego, CA, USA).

## 3. Results and Discussion

### 3.1. Extraction Yield, Total Phenolic, and Total Flavonoid Contents 

Table 1 shows the extraction yield and the contents of total phenolic (TPC) and total flavonoid (TFC) of *T. capitatus* extracts (TcLME, TcLAE, and TcLEO) obtained from the leaves powder formulations.

The findings result of extraction yield showed that the value of acetonic extract (around 30%) was more than double in comparison with the methanolic extract. Although the extraction yield for TcLEO was 3.15 ± 0.28, the result demonstrated an important yield when compared with other studies, where the values were 2.05%, 2.9%, 1.2%, and 2%, respectively [44,55,56,57]; this variation could be due of the location region, the pedoclimatic factors and sometimes to the extraction equipment used [58]. The very strong and aromatic odor with the yellow of TcLEO (Table 1) and its organoleptic parameters are in agreement with those listed in AFNOR standards [59].

The determination of TPC and TFC was done using a UV-visible spectrophotometer. The TPC in TcLME was observed to be significantly higher than TcLAE and TcLEO, with the values of 73 and 20.45 mg GAE/g of DE and 5.18 mg GAE/g of EO, respectively. Similar to TPC, TcLME was given a more important TFC than TcLAE and TcLEO, with the values of 13.15 and 6.79 mg CE/g of DE and 0.12 mg CE/g of EO, respectively.

The TPC in our TcLME (73 mg GAE/g of DE = 8.18 mg GAE/g of DW) was significantly lower than the ones from the different bioclimatic zone (sub-humid, upper semi-arid, means semi-arid, lower semi-arid, upper arid and lower arid) in Tunisia with the values were ranged from 12.6 to 42.7 mg GAE/g of DW [60]. Using aerial parts of *T. capitatus*, Msaada et al. [46] were found a lower amount of TPC in methanolic extract in four Tunisian regions, namely Kef, Grombelia, Bizerte, and Boukornine, with the values of 18.40, 12.40, 10.92, and 8.44 mg EAG/g of DE, respectively. However, Achour et al. reported a higher quantity of TPC (168.05 mg EAG/g of DE) than our TcLME. Concerning TFC, the previous study by Msaada et al. reported a higher amount in the four regions of the study (values of 63.64, 63.26, 46.31, and 26.83 mg CE/g DE, respectively) in comparison with our finding in TcLME.

To our knowledge, there has been no investigation on phenols contents of acetonic extract of *T. capitatus*, whereas TPC of EO was only reported in Morocco by Ouadan et al. [61] using a mix of leaves and flowers, and the value (4.74 mg GAE/mL of EO) was close to the amount given in our TcLEO (5.18 mg GAE/mL of EO).

Overall, these results indicated that methanol serves as a good extraction solvent of phenolic compounds from *T. capitatus*. Moreover, several studies have indicated that methanol and acetone are among the most practical solvents for the extraction of phenolic compounds [62].

### 3.2. Phytochemical Profiling by LC-MS of Extracts

Table 2 presented the characterization of the phenolic compounds by LC-MS of leaves of *T. capitatus* methanolic and acetonic extracts. The chromatographic separation demonstrated the presence of the nine phenolic compounds tested in both methanolic (Figure 1) and acetonic (Figure 2) extract. Paraben acid was found to be the major compound with the value of 188.77 and 126.04 µg/g of dry extract, followed by cinnamic acid with the value of 116.39 and 47.40 µg/g, and *p*-hydroxybenzoic acid with the value of 84.02 and 40.94 µg/g for methanolic and acetonic extracts, respectively. Similar to our results, the compounds syringic acid, *p*-coumaric acid, ferulic acid, and trans-cinnamic acid were determined in two previous studies from Tunisia [35,46]. These last references also identified other phenolic compounds, such as caffeic acid, chlorogenic acid, and rosmarinic acid [46], which were not analyzed in our study.

### 3.3. GC-MS Analysis of Thymus capitatus Essential Oil

The GC-MS chromatogram of TcLEO has recorded a total of 10 peaks (Figure 3) corresponding to 10 biomolecules (Table 3) that were recognized by relating their peak retention time, peak area (%), and mass spectral fragmentation patterns to that of the known compounds described by the NIST2014 library. The confirmation of the identified peaks was based on their retention indices relative to alkanes with those of authentic compounds and comparison with the retention indices values in the NIST online database of the corresponding compounds. Using these approaches allowed us to identify up to 99.5% of the total of the TcLEO. Carvacrol (79.22%) was the main compound identified in TcLEO, far ahead of o-cymene (5.12%), α-terpinolene (4.33%), caryophyllene (4.31%), and α-pinene (1.75%) (Figure 3, Table 3). Carvacrol is a phenol that is responsible for many diverse biological activities, including antimicrobial, antifungal, antitumor, antimutagenic, antigenotoxic, analgesic, antispasmodic, anti-inflammatory, angiogenic, antiparasitic, insecticidal, and hepatoprotective activities, and more others [63,64,65]. Previously, a study on TcLEO from the same collected region (Tetouan) was observed to be dominated by carvacrol (68.63%), *p*-cymene (4.84%), and γ-terpinene (3.75%), which were close to our findings [66]. According to our results, several studies from the same collection area using aerial parts of *T. capitatus* (leaves, tigers, and/or flowers) were reported that carvacrol was the main compound detected in their EO, with a percentage that varied from one study to another (65.96–92.27%) [67,68,69,70]. In contrast, a study noted that thymol was the major constituent identified in TcEO (aerial parts) from two areas in Tetouan Provence (Beni Aarouss 63% and Tamzzaght 55.85%) [71]. Another research from the same region of northern Morocco (Al Hoceima Provence) reported that *p*-cymene (18.9%) was the main compound identified in TcLEO, followed by carvacrol (13.4%), geranyl acetate (12.2%) and borneol (10.2%) [28].

Comparing our TcLEO results to ones from different Mediterranean countries, various studies were detected carvacrol as a predominated compound with the value of 58.66 to 81.49% in Tunisia [31,72,73,74], 72.8% in Italy [75], 69.83% in Spain [76] and 35.6% in Turkey [41]. In contrast, two studies reported that thymol was the most abundant constituent in TcLEO from Algeria and Tunisia, with a percentage of 51.22% and 89.06%, respectively [44,77]. Moreover, the most works on *T. capitatus* EO of aerial parts from different Mediterranean countries, including Algeria [30], Tunisia [45,57,78,79,80,81], Libya [82,83,84], Turkey [43], Greece [85], Italy [86,87,88], and Portugal [56,89,90] were found carvacrol as the main compound detected by GC-MS which is according to our TcLEO results.

Overall, the comparison of our results and literature ones were shown that the chemical composition of *T. capitatus* EO could be influenced by various factors such as growth condition, collection time, growth stage, plant parts, drying period, and other factors.

### 3.4. Antioxidant Activity by ORAC

The data is presented in Figure 4. Revealed the measured antioxidant activity of the methanolic, acetonic extracts, and EO of *T. capitatus* by ORAC assay. The results demonstrate that the highest antioxidant assay was observed in the TcLEO and TcLAE (about 5347 µmol Trolox Equivalent (TE)/mg of the sample), and the lowest antioxidant activity was observed in TcLME (400 µmol TE/mg of extract).

The highest activity observed in TcLEO may be related to carvacrol (the dominated compound detected in our chemical analysis) and may act as an antioxidant by giving hydrogen. Miguel et al. [90] found that the EO of *Thymus caespititius* and *Thymbra capitata* with the greatest antioxidant activity prevailed by the thymol and carvacrol. In addition, Aazza et al. [89] measured the antioxidant capacity by ORAC assay of six EO, and they found that the EO of *T. caespititius* and *T. capitata* had a significant ability with values of 2152.6 µmol TE/g and 1056 µmol TE/g, respectively. Our TcLME presents a great paradox; on the one hand, it is the extract richest in phenolic compounds and flavonoids (73 ± 6.157 µg GAE/mg of extract and 13.15 ± 0.517 µg CE/mg of extract, respectively). It also has the lowest value in comparison with TcLEO and TcLAE. On the other hand, it is the only extract that has shown prooxidant properties. This is not really surprising since phenolic compounds generally act as antioxidants at low concentrations, but at high concentrations, they can become prooxidants by their autooxidation, leading to the formation of semiquinone and the superoxide radical [91,92]. Furthermore, Silva et al. demonstrated that ascorbic acid, as known as a strong antioxidant, can also act as a prooxidant agent at certain concentrations while can cause an important degradation of the DNA bands [53]. Some studies indicated no correlations between the total phenolic content and the radical scavenging activity [93].

### 3.5. Assessment of DNA Degradation by T. capitatus Extracts

As the results of the ORAC assays suggested important antioxidant capacity, the next step was to analyze if the TcLAE, TcLME, and TcLEO were able to prevent DNA degradation, as the compounds phenolic can be both good protectors against DNA oxidation and they have prooxidant activity. This type of test has never been used when studying extracts of *T. capitatus*.

In Figure 5 (AO without iron), the protective effect of DNA degradation can be observed for concentrations ranging from 300 to 400 µg/mL for the methanolic extract; complete inhibition of DNA (100%) degradation was not observed. Moreover, the results observed the acetonic and EO oil showed a small to no protective effect. On the other hand, when DNA degradation was induced by the H_2_O_2_/iron system, the concentration of 200 µg/mL was capable of fully protecting DNA (100%) for methanolic and acetonic extracts Figure 5 (AO with iron) in comparison with EO. In contrast, in certain cases, inhibitions of desoxyribonucleic acid (DNA) degradation seem on top of 100%, indicating a greater fluorescence intensity of the band when exposure to the compounds, which can suggest that some interactions between the compound and desoxyribonucleic acid appear.

As can be seen in Figure 5 (PO without iron), no prooxidant effect was observed in the acetonic extract. The results revealed no DNA degradation for concentrations ranging from 150 to 400 mg/mL (negative values for DNA band degradation). However, for the methanolic extract and EO, there was important degradation of the DNA bands, which proved the prooxidant effect. The same thing occurred for the prooxidant activity with iron; all the extracts tested degraded the DNA bands (PO with iron).

### 3.6. Genotoxicity Activity

The extracts were evaluated with or without S9 fraction in order to recognize if metabolic activation could lead to mutagenicity and consequently be used for food applications. The results were considered positive or negative according to the folding rule, using a 2.5-fold, compared with the respective solvent control. The test was performed by adding 100 µg/mL of each plant extract to strain *Salmonella typhimurium* (TA98) with and without metabolic activation (S9). The numbers of revertant colonies per plate are listed in Table 4. Therefore, using the 2.5-fold rule to determine the presence or absence of mutagenicity: for samples diluted in DMSO 4% or aqueous solvent, values below 23 and 29 colonies, respectively, were considered non-mutagenic without metabolic activation (−S9); in the case of metabolic activation (+S9), samples with values below 41.25 and 42.5 colonies per plate in an aqueous solvent and DMSO 4%, respectively, were considered non-mutagenic. Thus, the results presented in Table 4 show that both the organic extracts (methanolic and acetonic) and the EO of *Thymus capitatus* were devoid of any mutagenic activity.

### 3.7. Antifungal Activity

The MIC and MFC for TcLME, TcLAE, and TcLEO against five fungal strains are presented in Table 5.

The TcLEO was able to inhibit the growth of all fungal strains studied at low concentrations (MIC 0.73–2.5 µL/mL and MFC 0.84–6.66 μL/mL); the lowest inhibition dose was observed for *M. canis* and the highest dose for *A. fumigatus*.

The TcLAE and TcLME were also able to inhibit the growth of all fungi. However, MIC and MFC were lower for TcLAE (MIC 0.52–1.25 mg/mL and MFC 0.63–2.08 mg/mL than for TcLME (MIC 0.73–2.5 mg/mL and MFC 1.10–9.17 mg/mL). Dermatophytes were most susceptible, being *M. canis* the most susceptible for the three tested extracts.

Comparing our results with the antifungal activity of *Lactarius sanguifluus* methanolic extract against the same fungal strains studied [51], it was found that MIC and MFC of *T. capitatus* are better.

Other works [94] proved that the aqueous, ethanolic extracts and EO of *T. capitatus* possess a strong antifungal and antibacterial effect. The maximum diameter of the inhibition zone (20 to 30.8 mm) against *C. albicans* and *M. canis* was reached at 50 µg/mL and 200 mg/mL concentrations, respectively, of EOs and aqueous extract [63,65,95,96].

Tabti et al. assessed the antifungal activity of the ethanol, hexane extracts, and EO from the aerial part of *T. capitatus* [19]. The EO was active against *Aspergillus niger*, *Aspergillus oryzae*, *Penicillium digitatum*, and *Fusarium solani* with a 100% inhibitory concentration between 0.1 and 0.5 μg/mL. Although, ethanol and hexane extracts had the lower activity with a minimum concentration causing 100% inhibition of mycelial growth higher than 1200 μg/mL.

According to previous studies, the antifungal activity of some *Thymus* EO was explained by the high content of phenol compounds (carvacrol and thymol) [97]. The study carried out by Maissa and Walid showed that the EO prepared from *T. capitatus* rich in carvacrol has an interesting effect compared with EO rich in thymol [98]. Many studies proved that the EO of *T. capitatus* was responsible for the antifungal activity of the plant [36,73,78,79,83].

The importance of phenolic hydroxyl groups for the antimicrobial activity of monoterpenoids has previously been reported [77,99,100]. This significant bioactivity of the extracts and EO of *T. capitatus* against the five fungal species (*T. rubrum*, *E. floccosum*, *M. canis*, *C. albicans*, and *A. fumigatus*) could be associated with their high content of phenolic terpene (carvacrol), and the synergistic interactions between different minor components.

## 4. Conclusions

The present study has been devoted to determining the total phenolics and flavonoids contents, identification of phenolic compounds by LC-MS, determination of the chemical composition of the EO, and evaluation of biological assays, including the antioxidant activity (ORAC and DNA protection), antifungal activity, and genotoxicity test of methanolic, acetonic extracts, and EO of *Thymus capitatus* plant leaves, an aromatic and medicinal plant in the north of Morocco, for its valorization with the objective of great exploitation. The chemical study on the EO of *Thymus capitatus* showed the presence of a majority of compounds such as carvacrol, o-cymene, and α-terpinolene, besides other components. In addition, the chromatographic analysis by LC-MS of methanolic and acetonic extracts revealed that phenolic compounds paraben acid, cinnamic acid, and *p*-hydroxybenzoic acid were the major compounds. The determination of TPC and TFC has shown that methanolic extract contains the highest content in comparison with acetonic extract and EO. Besides, the results of biological assays (antioxidant, antifungal activity, and genotoxicity test) were globally very significant.

These results suggest that the extracts and essential oil of the *Thymus capitatus* plant can present an interesting alternative natural, which can be useful for food and pharmaceutical treatment.

## Figures and Tables

**Figure 1 antibiotics-11-00681-f001:**
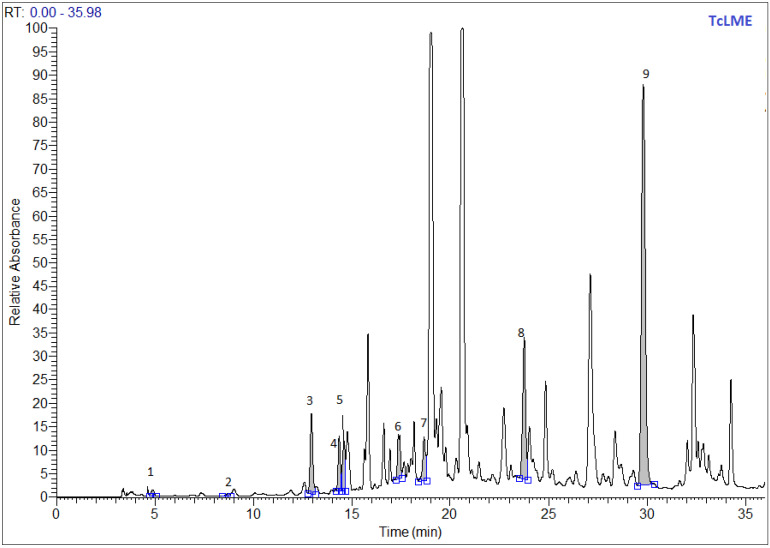
LC–MS chromatogram of phenolic compounds in the methanolic extracts of *Thymus capitatus* leaves, as detected at 280 nm.

**Figure 2 antibiotics-11-00681-f002:**
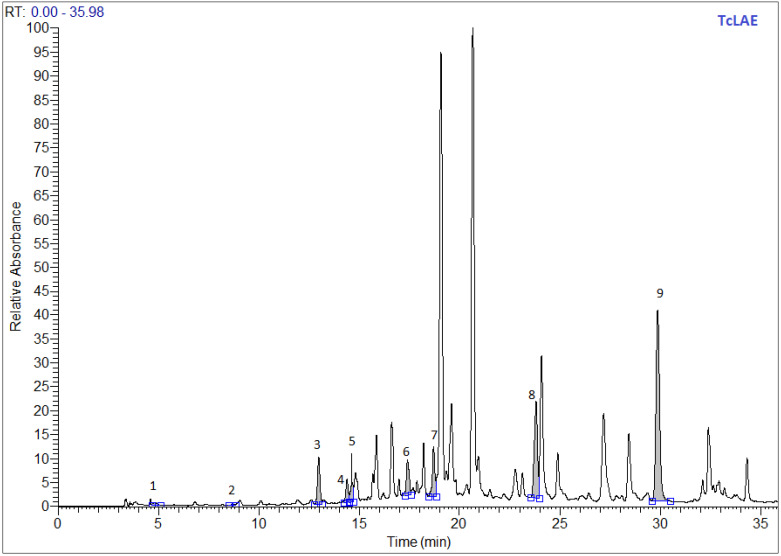
LC–MS chromatogram of phenolic compounds in the acetonic extracts of *Thymus capitatus*, as leaves detected at 280 nm.

**Figure 3 antibiotics-11-00681-f003:**
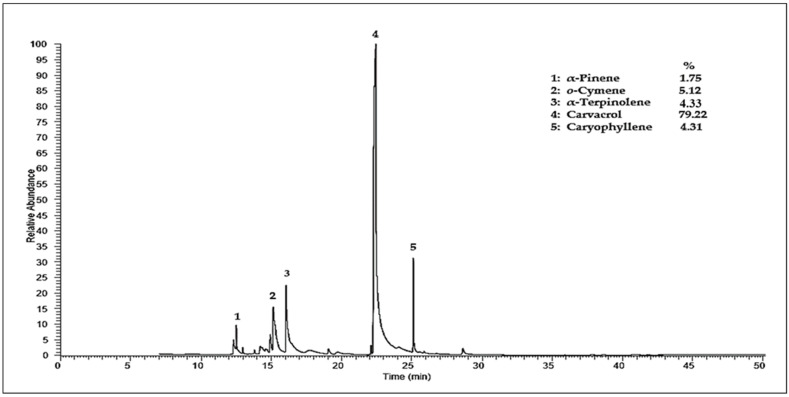
Chromatographic profile by GC-MS of *Thymus capitatus* essential oil leaves.

**Figure 4 antibiotics-11-00681-f004:**
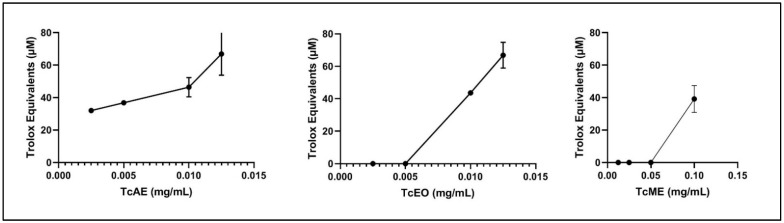
Antioxidant activity of methanolic, acetonic extracts, and essential oil of *Thymus capitatus* by the ORAC method.

**Figure 5 antibiotics-11-00681-f005:**
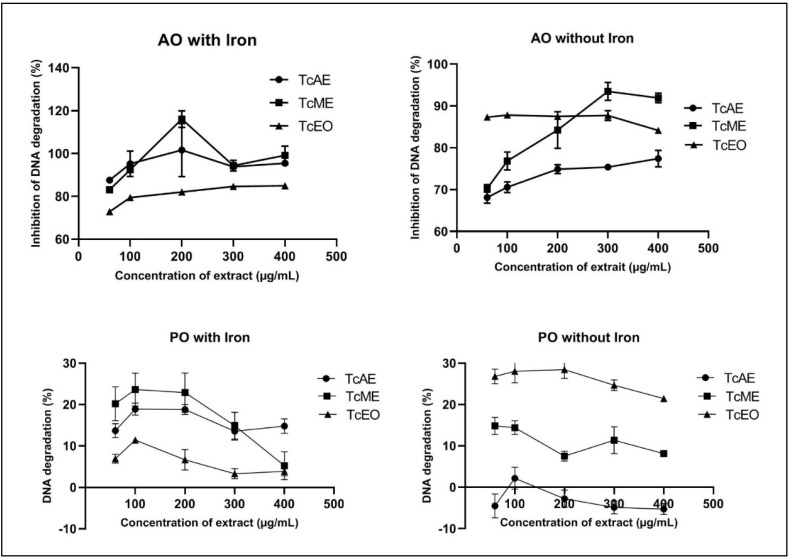
Assessment of inhibition DNA degradation and DNA degradation in AO and PO assays at different concentrations of *Thymus capitatus* extracts.

**Table 1 antibiotics-11-00681-t001:** Yield extraction (%) and Bioactive Compounds (total phenolics and flavonoids) in the methanolic extract (TcLME), acetonic extract (TcLAE), and essential oil (TcLEO) of *Thymus capitatus* leaves.

Extractive Yield and Bioactive Compounds	Plant Extracts		
	TcLME	TcLAE	TcLEO
Extractive Yield (%)	11.2 ± 1.79	29.34 ± 4.11	3.15 ± 0.28
Total phenolic contents (mg GAE/g of DE or EO)	73.00 ± 6.18	20.45 ± 2.18	5.18 ± 0.68
Total flavonoid contents (mg CE/g of DE or EO)	13.15 ± 0.52	6.79 ± 0.37	0.12 ± 0.01

**Table 2 antibiotics-11-00681-t002:** Quantification of phenolic compounds in the methanolic (TcLME) and acetonic (TcLAE) extracts of *Thymus capitatus* leaves (µg/g of DE).

Peak	Phenolic Compounds	TcLME	TcLAE
1	Gallic acid	6.48	3.76
2	Protocatechuic acid	4.04	1.09
3	*p*-hydroxybenzoic acid	84.15	43.68
4	Vanillic acid	45.43	18.22
5	Syringic acid	25.87	9.77
6	*p*-coumaric acid	15.60	8.49
7	Ferulic acid	27.39	23.15
8	Paraben acid	188.77	126.04
9	Cinnamic acid	116.39	47.40

**Table 3 antibiotics-11-00681-t003:** Chemical composition of *Thymus capitatus* essential oil leaves (percentage—%, retention time-RT, and retention indexes-RI).

Chemical Compound	RT	RI	%
β-Thujene	12.30	922	1.10
α-Pinene	12.50	945	1.75
Camphene	12.96	955	0.38
Pseudolimonene	14.21	1003	0.98
α-Terpinene	14.90	1019	1.50
o-Cymene	15.12	1027	5.12
α-Terpinolene	16.02	1088	4.33
Carvacrol	22.44	1298	79.22
Caryophyllene	25.10	1418	4.31
Caryophyllene oxide	28.63	1582	0.81
Total			99.5

**Table 4 antibiotics-11-00681-t004:** Genotoxicity of *Thymus capitatus* extracts and essential oil against *Salmonella Typhimurium* TA98 (+S9) and bacterial strain TA98 (−S9).

Extracts Solvent	Dose/Plate (µg/mL)	TA98 (+S9)	TA98 (−S9)
TcLME	100	17.50 ± 3.50	16.50 ± 03.5
TcLAE	100	14.00 ± 01	19.50 ± 00.5
TcLEO	100	6.00 ± 00	00.00
DMSO 4%	100	9.00 ± 01	17.00 ± 01
Water	100	11.5 ± 0.50	16.50 ± 1.5
2-Aminoanthracen	10	1232 ± 28	-
Daunomycin	06	-	481.50 ± 3.5

**Table 5 antibiotics-11-00681-t005:** Minimum inhibitory concentration (MIC) and minimum fungicidal concentration (MFC) of the *Thymus capitatus* extracts (TcLME, TcLAE), and essential oil (TcLEO).

Fungal Strains	TcLEO (µL/mL)		TcLAE (mg/mL)		TcLME (mg/mL)	
	MIC	MFC	MIC	MFC	MIC	MFC
*C. albicans*	2.50 ± 0.00	3.75 ± 1.44	0.63 ± 0.00	2.08 ± 0.54	2.5 ± 0.00	3.75 ± 1.37
*A. fumigatus*	2.25 ± 0.56	6.66 ± 2.74	1.25 ± 0.00	1.88 ± 0.68	2.5 ± 0.00	9.17 ± 2.04
*T. rubrum*	1.46 ± 0.95	1.67 ± 0.72	0.58 ± 0.13	0.94 ± 0.36	1.15 ± 0.25	2.5 ± 0.00
*E. floccosum*	1.46 ± 0.95	2.08 ± 0.72	0.63 ± 0.00	1.25 ± 0.00	1.25 ± 0.00	2.5 ± 0.00
*M. canis*	0.73 ± 0.47	0.84 ± 0.36	0.52 ± 0.17	0.63 ± 0.00	0.73 ± 0.25	1.10 ± 0.31

The results are presented as mean ± SD (*n* = 6). *Candida albicans*, *C. albicans*; *Aspergillus fumigatus*, *A. fumigatus*; *Trichophyton rubrum*, *T. rubrum*; *Epidermophyton floccosum*, *E. floccosum*; *Microsporum canis*, *M. canis*.

## Data Availability

Not applicable.

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
