# Peer review of "Phytochemical Composition, Antioxidant and Antifungal Activity of Thymus capitatus, a Medicinal Plant Collected from Northern Morocco"

_antibiotics, 2022, doi:10.3390/antibiotics11050681_

Round 1
Reviewer 1 Report
It is well documented that Thymus capitatus is an endemic medicinal plant to the Mediterranean region and has traditionally been considered as an anthelmintic, antispasmodic, carminative, emmenagogue, expectorant, rubefactient, sedative, stimulant, and tonic. Its total phenolic content and total flavonoid content had been determined (Journal of Functional Foods, 2012, 4 (3):661-669; International Journal of Food Properties, 2016, 19(6):1381-1390) and its components in essential oil of T. capitatus from Morocco was analyzed using GC-MS method (Journal of Medicinal Plants Research, 2011, 5(24):5773-8). Therefore, the present work seems to lack enough novelty except genotoxicity assay.
Author Response
The responses are attached

Reviewer 2 Report
The manuscript "Phytochemical Composition, Antioxidant and Antifungal Activity of Thymus capitatus, a Medicinal Plant collected from Northern Morocco". The paper presents novel informations to the journal "Antobiotics" and for the VSI 'Recent Advances in Antimicrobial Drug Discovery'. Therefore, should be accepted after Major Review. Below, there are several comments to improve the scientific quality of the manuscript.
The abstract should be a total of about 200 words maximum.
Line 61. The wide variety of purposes should de described and included in the sentence.
Line 82-84: There are also the carotenoids, which present antioxidant activity. Please include.
Line 94-100: Here it will be interesting to include that the essential oil can be (nano)encapsulated with zein to increase the shelf life of food by inhibiting fungus.
Line 101-105: Avoid use abbreviation in the first time in the aim. For instance, LC-MS and ORAC.
Line 141, 154, 161, 170, 248: Plese correct the reference Erbiai et al. (2021b). The year should not be used. Also, avoid self-citation. El Hadi Erbiai is a co-author of the present study. If possible, include the original methodology. Erbiai et al. (2021b) developed the method or just used?
Line 194-195: "activity was performed according to the procedure employed by the authors [45]...". What authors? For [45] or from the present study? Not clear...
Line 231-233: This is not the correct way to use/write the equation. Please, revise it.
Line 276: "the result demonstrated an important yield when compared with other studies such as M. Ajjouri in Morocco, Faleiro et al. in Portugal, Mkaddem et al., and Hedhili et...". Why to write the name of the authors? It is not necessary. Revise.
Table 1. In my opinion, the color, appearance and odor of the extract/EO, is not necessary to desribe, since these characteristics are expected. That is, an EO will be very strong aromatic, if not, it is not EO.
Line 298, 302, 306, 377, 379 (and many other along the paper!): Correct the reference (should not present the year).
Table 3: I would like to see the chromatogram in the paper.
Figure 1 should be improved to better observe the chromatogram lines.
Figure 2. is not necessary. What is the total antioxidant capacity by ORAC? Just a number is necessary. I don't see the relevance of this Figure.
It will be interesting to include pictures of the disk diffusion assay against the tested microorganisms. This will corroborate the data presented in Table 6.
Author Response
The responses are attached.

Reviewer 3 Report
The authors in the manuscript titled “Phytochemical Composition, Antioxidant and Antifungal Activity of Thymus capitatus, a Medicinal Plant collected from Northern Morocco”, have described the methods to determine Total Phenolic Contents (TPC) and Total Flavonoid Contents (TFC) in T. capitatus, a Mediterranean endemic medicinal plant from northern morracco. In addition, they also reported the chemical composition of the essential oils from T. capitatus and their antioxidant, antifungal activities. To the best of my knowledge, I believe the manuscript is of very good quality and needs no major changes, since it is well structured, clear and well organized, conceptually and methodologically.
I have a couple of corrections,
- Page 1, Line 36. Change ORAG to ORAC.
- Page 3, Line 143. Please correct Whatman N° 4 Filter paper.
Author Response
The responses are attached.

Round 2
Reviewer 1 Report
It seems that all techniques used in this work are very common and the research project except antifungal tests has no relationship with Antibiotics.
Reviewer 2 Report
The authors adressed all the suggested points. The paper can be now accepted for publication.